# Membrane Fouling Diagnosis of Membrane Components Based on MOJS-ADBN

**DOI:** 10.3390/membranes12090843

**Published:** 2022-08-29

**Authors:** Yaoke Shi, Zhiwen Wang, Xianjun Du, Bin Gong, Yanrong Lu, Long Li, Guobi Ling

**Affiliations:** 1College of Electrical and Information Engineering, Lanzhou University of Technology, Lanzhou 730050, China; 2Key Laboratory of Gansu Advanced Control for Industrial Processes, Lanzhou University of Technology, Lanzhou 730050, China; 3National Demonstration Center for Experimental Electrical and Control Engineering Education, Lanzhou University of Technology, Lanzhou 730050, China; 4GS-Unis Intelligent Transportation System & Control Technology Co., Ltd., Lanzhou 730050, China

**Keywords:** adaptive learning rate, MOJS, DBN, stability proof, membrane fouling diagnosis

## Abstract

Given the strong nonlinearity and large time-varying characteristics of membrane component fouling in the membrane water treatment process, a membrane component-membrane fouling diagnosis method based on the multi-objective jellyfish search adaptive deep belief network (MOJS-ADBN) is proposed. Firstly, the adaptive learning rate is introduced into the unsupervised pre-training phase of DBN to improve the convergence speed of the network. Secondly, the MOJS method is used to replace the gradient-based layer-by-layer weight fine-tuning method in traditional DBN to improve the ability of network feature extraction. At the same time, the convergence of the MOJS-ADBN learning process is proven by constructing the Lyapunov function. Finally, MOJS-ADBN is used in the membrane packaging diagnosis to verify the performance of the model diagnosis. The experimental results show that MOJS-ADBN has a fast convergence speed and a high diagnostic accuracy, and can provide a theoretical basis for membrane fouling diagnosis in the actual operation of membrane water treatment.

## 1. Introduction

Membrane bioreactor (MBR) technology, as an important means in sewage treatment engineering, is a new wastewater treatment process that combines membrane technology and biological treatment technology and is mainly composed of membrane components and the bioreactor [1,2,3]. It has been recognized as one of the most promising new technologies in the field of water treatment in the 21st century due to its excellent comprehensive performance. However, membrane fouling of membrane components will increase the operating cost of the MBR, becoming a bottleneck problem that restricts its wide application [4,5]. Therefore, researchers are gradually focusing on membrane component–membrane fouling diagnosis technology in the field of water treatment. The traditional fault diagnosis method is divided into three steps. Firstly, the signal is preprocessed by denoising and decomposing. Secondly, the preprocessed signal can obtain its time domain, frequency domain, or other features through certain feature extraction methods. The feature extraction methods include wavelet transform [6], synchronous extraction [7], empirical wavelet transform [8], and so on. These methods filter the useless features of the signal, making the desired fault features more obvious. Finally, the extracted features are input into the classifier based on machine learning for training; the classification of faults can be recognized by the training classifier. The backpropagation neural network (BP-NN) [9] and support vector machine (SVM) [10] have been applied to fault classification. The above methods have the characteristics of simple feature extraction and easy adjustment of classifier parameters, and the final diagnostic recognition rate can meet most requirements. However, the above methods still separate fault feature extraction and diagnosis and recognition, which require a lot of expert experience in signal feature processing and rely on the ability of manual feature extraction, which is limited [11,12,13]. Similarly, the traditional fault diagnosis method based on signal processing adopts the manual extraction of features and input of the classification model for fault identification [14,15,16]. The process relies heavily on manual experience and prior knowledge, which are insufficient in large data scales and fast acquisition speeds. 

In view of the dynamic and nonlinear characteristics of the membrane water treatment system, the traditional diagnostic model is inefficient, and the potential valuable features are ignored in the offline modeling stage, resulting in false alarms and inaccurate interpolation [17]. As a breakthrough in the field of modern artificial intelligence, deep learning can automatically learn valuable features from original feature sets and even original data, which means that deep learning can largely get rid of the dependence on advanced signal processing technology, artificial feature extraction, and cumbersome feature selection technology. Therefore, deep learning is widely used in the field of fault diagnosis with its powerful learning ability and feature extraction ability [18,19,20]. Ba-Alawi et al. proposed an inclusive framework for missing data interpolation and sensor self-verification based on the variational automatic encoder and deep residual network structure integration [21]. By learning the potential probability distribution of input data, complex features are automatically extracted to reduce the risk of gradient disappearance. By inputting missing data, detecting anomalies, identifying fault sources, and reconstructing fault data to a normal state, the reliability of fault sensors is improved. In recent years, a series of deep learning fault diagnosis models based on the convolutional neural network (CNN) have been greatly improved in diagnosis efficiency and accuracy. Shi et al. used attention mechanisms and improved convolutional neural networks to diagnose membrane pollution, which improved diagnostic accuracy and efficiency [22,23]. However, the deep learning model requires a large number of data to optimize parameters and is prone to over-fitting [24,25]. More researchers have studied the application of deep belief networks (DBNs) in the field of fault diagnosis. DBNs have strong feature extraction abilities, which can automatically extract features from a large number of data, reduce the dependence on expert fault diagnosis experience and signal processing technology, and reduce the uncertainty of feature extraction and fault diagnosis caused by manual participation in traditional methods [26,27,28]. A DBN characterizes the complex mapping relationship between signals and the health status by establishing a deep model, which is suitable for the diagnosis and analysis of diverse, nonlinear, high-dimensional health monitoring data in the context of big data [29]. Therefore, applying a DBN to the field of fault diagnosis has certain timeliness, practicality, and versatility. Zhao et al. proposed a fault diagnosis method based on a DBN, which adaptively extracted features from the original time series signals, increasing flexibility [30]. Simulation results show the effectiveness of this method in fault diagnosis. The structural parameters of a typical DBN model are determined by the learning rate [31]. Therefore, Liu et al. applied an optimized DBN to improve the accuracy of fault diagnosis [32]. Zhang et al. proposed a fault diagnosis model of complex chemical processes based on an extensible DBN [33]. With the help of mutual information technology, a DB subnetwork is used to extract individual fault features in the space–time domain. A global two-layer backpropagation network has been trained and used for fault classification, and the effect of fault diagnosis of this method was verified. Dai proposed a DBN fault diagnosis model with an improved model structure, which adopted multi-layer and multi-dimensional mapping to extract more detailed fault type differences and accurately diagnose faults [34]. Zhu et al. introduced a DBN network into a multi-sensor information fusion model to identify uncertain, unknown, and changing fault modes [35]. Compared with the traditional artificial neural network information fusion diagnosis method, this method has higher recognition accuracy. Su et al. used the model after GWO optimized the parameters of the support vector machine to diagnose the signal features extracted by DBN, realizing the online detection of equipment faults, and improving the diagnostic accuracy [36]. Zhu proposed an intelligent fault diagnosis method based on PCA and DBN [37]. The PCA method is used to reduce the dimension of the original signal, to extract fault eigenvalues and eigenvectors. The modified samples are then trained and tested by DBN for fault classification and diagnosis. This method does not need complex signal processing of the original data, so it is easy to implement and has wide applicability. Due to the uncertainty of the dynamic system model of membrane water treatment, the nonlinearity of data signals, and the uncertainty of the membrane fouling state, the extraction of membrane fouling characteristics from membrane components is in trouble. In addition, with the increase in the scale and complexity of industrial control systems, membrane fouling data signals are often composed of a large number of high-dimensional data, which makes the processing of original membrane fouling data more complex. 

Based on the above problems, this article proposes a membrane-packing diagnosis method based on MOJS-ADBN to optimize the DBN from the perspective of unsupervised learning and supervised learning: we used an adaptive learning rate to accelerate network convergence, and prove that the unsupervised part optimized by the adaptive learning rate is stable. The supervised part uses the MOJS algorithm optimization to fine-tune the weight and proves that MOJS optimization has global convergence and stability in the Lyapunov meaning. We used the MOJS-ADBN model as an example of the membrane fouling diagnosis of the parallel ultrafiltration membrane component and verified the comprehensive performance of the MOJS-ADBN model through a number of comparative tests.

## 2. Traditional DBN Model

### 2.1. Subsection

In 2006, Hinton proposed a DBN, which is a probability generation model composed of multiple restricted Boltzmann machine (RBM) stacks. 

As a two-layer network, a RBM is bidirectionally connected by the visible layer and the hidden layer, and the neurons of the same layer network are independent of each other. The visual layer is used to input training data, while the hidden layer is used to extract features. The structure diagram of the RBM is shown in Figure 1. In the formula, $w_{1}^{R}$ represents the connection weight, b is the bias coefficient of the hidden layer, and a is the bias coefficient of the visible layer.

The feature extraction process of the DBN is classified into two stages: the pre-training stage and fine-tuning stage. In the pre-training stage, all RBMs are first pre-trained layer-by-layer, unsupervised, to form a feature model of unsupervised learning. Next, the supervised algorithm is used for reverse training, and all the initial connection weights of RBM are fine-tuned, to reduce the error caused by training, which is conducive to the DBN to extract the essential characteristics of the input data. The structure is shown in Figure 2.

### 2.2. Unsupervised Learning

To determine the initial weight of the network, Hinton used an unsupervised training method to learn the parameters. One RBM includes a visible layer and a hidden layer, which are represented by *v* and *h*, respectively. Given the model parameter $\theta =\{w^{R},a,b\}$, the joint probability distributions P(v,h;θ) of the visible layer and the hidden layer are defined by the energy function E(v,h;θ) as:(1)$$P\left(v,h;\theta \right)=\frac{1}{Z}e^{-E\left(v,h;\theta \right)}$$
(2)$$P\left(v;\theta \right)=\frac{1}{Z}\sum_{h}e^{-E\left(v,h;\theta \right)}$$

For an RBM with Bernoulli (visible layer) distribution–Bernoulli (hidden layer) distribution, the energy function of the unit joint configuration is defined as:(3)$$E\left(v,h\right)=-\sum_{i=1}^{m}\sum_{j=1}^{n}v_{i}w_{ij}^{R}h_{j}-\sum_{i=1}^{m}a_{i}v_{i}-\sum_{j=1}^{n}b_{j}h_{j}$$

In the formula, $w_{ij}^{R}$ is the connection weight of RBM, *a_i_* and *b_j_* are the offsets of the visible layer cells and hidden layer cells, respectively.

The conditional distributions of *v* and *h* are:(4)$$P\left(h_{j}=1/v;\theta \right)=\sigma \left(b_{j}+\sum_{i=1}^{m}v_{i}w_{ij}^{R}\right)$$
(5)$$P\left(v_{i}=1/h;\theta \right)=\sigma \left(a_{i}+\sum_{j=1}^{n}w_{ij}^{R}h_{j}\right)$$

In the formula, *σ* is the activation function.

The probability value standard of the visible layer and the hidden layer is usually achieved by setting a threshold, because the visible layer and the hidden layer are Bernoulli binary states. Taking the hidden layer as an example, it can be expressed as:(6)$$h_{j}\left\{ \begin{array}{l} 0\;if\;p\left(h_{j}=1/v\right)<\delta \\ 1\;if\;p\left(h_{j}=1/v\right)>\delta \end{array}\right.\;$$

In the formula, *δ* is a constant between 0.5 and 1. 

We calculate the gradient of the log-likelihood function lgP(v;θ), and the RBM weight update formula can be obtained as:(7)$$w_{ij}^{R}=w_{ij}^{R}+\eta \Delta w_{ij}^{R}$$
(8)$$\Delta w_{ij}^{R}=E_{\mathrm{data}}\left(v_{i}h_{j}\right)-E_{\mathrm{model}}\left(v_{i}h_{j}\right)$$

In the formula, *η* represents the learning rate, $E_{\mathrm{data}}\left(v_{i}h_{j}\right)$ are the data expectations observed in the training set, $E_{\mathrm{model}}\left(v_{i}h_{j}\right)$ is the expectation on the distribution determined by the model, and $E_{\mathrm{model}}\left(v_{i}h_{j}\right)$ can be obtained by the Gibbs approximation.

### 2.3. Supervised Learning

Supervised learning involves fine-tuning the weight *w^R^* obtained by unsupervised learning. Taking the output layer and the last hidden layer of Figure 2 as examples, let *F* be the expected output of the model and define the cross-entropy function as the loss function:(9)$$F=-\frac{1}{n}\sum_{i}\left[y_{i}\ln y_{i}^{\prime }+\left(1-y_{i}\right)\ln\left(1-y_{i}^{\prime }\right)\right]$$

In the formula, *y_i_* represents the output of the target output after SoftMax, $y_{i}^{\prime }$ represents the output of the expected output after SoftMax, and *n* represents the number of categories.

The weight update formula can be expressed as:(10)$$w_{out}\left(\tau +1\right)=w_{out}\left(\tau \right)-\eta \frac{\partial F\left(\tau \right)}{\partial w_{out}\left(\tau \right)}$$

Using this method, the weight $w=\left(w_{out},w_{l},w_{l-1},\cdots ,w_{2},w_{\mathrm{in}}\right)$ of the whole DBN network can be obtained by fine-tuning from the top output layer to the bottom input layer.

## 3. MOJS-ADBN Learning Algorithm

### 3.1. Adaptive Learning Rate CD Algorithm

In the unsupervised learning process of the DBN, Gibbs sampling, as the core of the contrastive divergence (CD) algorithm, is a Markov chain Monte Carlo (MCMC) algorithm. When it is difficult to directly sample the joint distribution, it is used to generate a set of approximate observations of a specific multi-parameter probability distribution. Gibbs sampling mainly consists of three steps.

(1) The Gibbs chain is initialized with sample *V* to obtain the visual layer input $v^{\left(0\right)}$.

(2) According to Formulas (4)–(6), sampling is carried out, respectively. In the formula, $h^{\left(t\right)}$ is obtained by sampling $P\left(h^{\left(t\right)}/v^{\left(t\right)};\theta \right)$, $v^{\left(t+1\right)}$ is obtained by sampling $P\left(v^{\left(t+1\right)}/h^{\left(t\right)};\theta \right)$, *t* is the number of sampling steps.

(3) We repeat the second stage.

Because each RBM requires multiple iterations, the fixed learning rate *η* is prone to convergence difficulties. Therefore, the adaptive learning rate is used to determine the learning rate *η* according to the updates direction of the parameters. The principle of the adaptive learning rate is that the learning rate will increase if the parameter update direction is the same after two consecutive iterations, and the learning rate will decrease if the parameter update direction is opposite after two consecutive iterations. The update mechanism of the adaptive learning rate *η* is as follows:(11)$$\eta =\left\{ \begin{array}{l} B\eta \;{\left(\Delta w_{ij}^{R}\right)}^{\left(t\right)}+\;{\left(\Delta w_{ij}^{R}\right)}^{\left(t+1\right)}=\;\left|{\left(\Delta w_{ij}^{R}\right)}^{\left(t\right)}\right|\;+\;\left|{\left(\Delta w_{ij}^{R}\right)}^{\left(t+1\right)}\right|\;\\ b\eta \;{\left(\Delta w_{ij}^{R}\right)}^{\left(t\right)}+\;{\left(\Delta w_{ij}^{R}\right)}^{\left(t+1\right)}<\;\left|{\left(\Delta w_{ij}^{R}\right)}^{\left(t\right)}\right|\;+\;\left|{\left(\Delta w_{ij}^{R}\right)}^{\left(t+1\right)}\right|\; \end{array}\right.\;$$
(12)$$\;{\left(\Delta w_{ij}^{R}\right)}^{\left(t\right)}=v_{i}^{\left(t\right)}h_{i}^{\left(t\right)}-v_{i}^{\left(t+1\right)}h_{i}^{\left(t+1\right)}$$
(13)$$\;{\left(\Delta w_{ij}^{R}\right)}^{\left(t+1\right)}=v_{i}^{\left(t+1\right)}h_{i}^{\left(t+1\right)}-v_{i}^{\left(t+2\right)}h_{i}^{\left(t+2\right)}$$

In the formula, *B* = 1.4, *b* = 0.7.

### 3.2. Supervised Fine Adjustment Based on MOJS

The three main features of MOJS are as follows: (1) Archiving is integrated into the jellyfish search to save and retrieve Pareto optimal solutions. (2) The crowding distance and roulette selection are used to effectively manage the archive population, including the optimal non-dominated solution in the spatial search process. (3) To alleviate local optimization, Lévy flight, an elite group, is added to MOJS based on opposite jumping. The weights obtained from the unsupervised process are fine-adjusted by MOJS.

#### 3.2.1. Time Control Function

Jellyfish are attracted by nutrients in the ocean current; they gather in the ocean current (and thus form jellyfish groups). There are also movements in jellyfish groups, namely passive movement (A-type movement) and active movement (B-type movement). The transformation of jellyfish (from A-type movement to B-type movement) is affected by the time control function c(*t*), and its expression is as follows:(14)$$c\left(t\right)=\left|\left(1-\frac{t}{Max_{iter}}\right)\times \left(2\times \mathrm{rand}\left(0,1\right)-1\right)\right|$$

In the formula, c_0_ = 0.5.

#### 3.2.2. Elite Choice

We added a file to the MOJS algorithm to store and retrieve the best approximation of the real Pareto optimal solution in the optimization process. The selection of elite targets was set in the area with the least jellyfish in the Pareto optimal frontier. The recognition method of this region involved dividing the search space by finding the best elite target and the worst target of the obtained Pareto optimal solution, defining a hypersphere and n grid elements covering all solutions, and dividing the hypersphere into equal sub-hyperspheres in each iteration; the roulette mechanism was used to select. The roulette mechanism can improve the distribution of the whole Pareto optimal frontier. When there are more Pareto optimal numbers, the probability of being selected is smaller, as shown in the following formula:(15)$$P_{i}=\frac{C}{N_{i}}$$

In the formula, *C* = 10, N_i_ is the number of Pareto optimal solutions obtained in segment *i*. 

#### 3.2.3. Lévy Flight

The behaviors of most flying animals can be described by Lévy flight when the spatial dimension of a random walk is higher than one dimension and the step size distribution of Lévy flight is isotropic; we used the Mantegna algorithm to generate a stable step size:(16)$$Lé\mathrm{vy}\left(s\right)∼s=\frac{u}{{\left|v\right|}^{\frac{1}{\tau }}},\;0<\tau \leq 2$$

In the formula, u and v obey normal distributions: $u∼N\left(0,\sigma_{u}^{2}\right),\;v∼N\left(0,\sigma_{v}^{2}\right)$.
(17)$$\sigma_{u}={\{\frac{\Gamma \left(1+\tau \right)\sin\left(\frac{\pi \tau }{2}\right)}{\Gamma \left[\frac{1+\tau }{2}\right]\tau 2^{\left(\tau -1\right)/2}}\}}^{\frac{1}{\tau }},\;\sigma_{v}=1,\;\tau =1.5\;$$

In the formula, Γ(z) is Gamma distribution: $\Gamma \left(z\right)=\int_{0}^{\infty }t^{z-1}e^{-t}\mathrm{dt}$.

#### 3.2.4. Update and Archive

In the iteration process, the archived file will be updated every time, and it may reach the upper limit of the total number in the optimization process. We used the management mechanism to filter the archived files, and the specific contents are as follows:

(1) If there is a solution that can play a leading role in the Pareto optimal solution in the original archive, we store the solution and delete the dominant solution in the original archive.

(2) If there is a solution A, and there is no dominant relationship between the original archived solution, the solution in the original archive will be retained. If the number of archives does not reach the upper limit, solution A will be added to the archive file.

(3) If the number of archives reaches the upper limit, the solution will be deleted from the stage with the most filling, and solution A will be added to the archive file.

(4) If there is a solution A that can be dominated by the original archiving solution, then we eliminate solution A. 

To effectively select solutions to be deleted from the archive, the worst (the most jellyfish) hypersphere should be selected to prevent jellyfish from searching in crowded areas without food. The selection method is realized through the roulette wheel mechanism, and the probability of each segment is:(18)$$P_{i^{\prime }}=\frac{N_{i}}{C}$$

In the formula, *C* = 10, N*_i_* is the number of Pareto optimal solutions obtained in segment *i*.

#### 3.2.5. MOJS

We used Lévy flight to speed up the local search along the ocean current; the formula of the ocean current motion is:(19)$$X_{i}\left(t+1\right){=\mathrm{EL}\_X}_{i}\left(t\right)+\vec{\mathrm{trend}}⊗Lé\mathrm{vy}\left(s\right)$$
(20)$$\vec{\mathrm{trend}}{=X}^{*}\left(t\right)-3\times \mathrm{rand}\left(0,1\right)\times \frac{\sum \mathrm{EL}\_X}{n_{\mathrm{pop}}}$$

In the formula, ${\mathrm{EL}\_X}_{i}\left(t\right)$ is the elite member in $X_{i}\left(t\right)$, ∑EL_X is the elite group, n_pop_ is the group size, $X^{*}\left(t\right)$ is the elite solution with time t selected in the archive. 

Similarly, we used elite solutions to replace the current best solutions of active movements and passive movements in jellyfish groups. 

Passive movement:(21)$$X_{i}\left(t+1\right){=X}^{*}\left(t\right)+\left({\mathrm{EL}\_X}_{i}\left(t\right){-X}^{*}\left(t\right)\right)⊗Lé\mathrm{vy}\left(s\right)$$

Active movement:(22)$$X_{i}\left(t+1\right){=X}^{*}\left(t\right)+\vec{\mathrm{Step}}$$

In the formula:(23)$$\vec{\mathrm{Step}}=\mathrm{rand}\left(0,1\right)\times \vec{\mathrm{Direction}}$$
(24)$$\vec{\mathrm{Direction}}=\left\{ \begin{array}{l} {\mathrm{EL}\_X}_{j}\left(t\right){-\mathrm{EL}\_X}_{i}\left(t\right)\;\mathrm{if}\;{\mathrm{EL}\_X}_{i}\left(t\right)≻{\mathrm{EL}\_X}_{j}\left(t\right)\\ {\mathrm{EL}\_X}_{i}\left(t\right){-\mathrm{EL}\_X}_{j}\left(t\right)\;\mathrm{if}\;{\mathrm{EL}\_X}_{j}\left(t\right)≻{\mathrm{EL}\_X}_{i}\left(t\right) \end{array}\right.$$

#### 3.2.6. Population Initialization

Logistic mapping, compared with the random initialization, is not easy to produce premature convergence and ensures population diversity. The formula is as follows:(25)$$X_{i+1}{=\eta X}_{i}\left({1-X}_{i}\right),\;0\leq X_{0}\leq 1$$

In the formula, X_i_ is the logistic value of the *i*-th jellyfish position, X_0_ is used to generate the initial population of jellyfish, η is equal to 4, $X_{0}\in \left(0,1\right),X_{0}\notin \{0.0,\;0.25,\;0.75,\;1.0\}$.

#### 3.2.7. Increase Diversity through Opposition-Based Jumping

This mechanism is effective when the population is approximately transformed into the optimal solution. If the jump condition $\mathrm{rand}\left(0,1\right)<\frac{t}{{\mathrm{Max}}_{\mathrm{iter}}}$ is satisfied, the corresponding population based on opposition $X\prime_{i}\left(t\right)$ is calculated and n_pop_ is calculated. After generating a new population through evolution, the most suitable individual is selected from the current population and the opposite population. In the formula, based on the opposite $X\prime_{i}\left(t\right)$ population, the calculation formula is:(26)$$X\prime_{i}\left(t\right)=\left({\mathrm{Lb}}_{i}{+\mathrm{Ub}}_{i}\right){-X}_{i}\left(t\right)$$

We extracted the hidden layer states obtained by unsupervised learning, and then carried out MOJS fine adjustment in sequence, according to the above steps, $w=\left(w_{out},w_{l},w_{l-1},\cdots ,w_{2},w_{2},w_{\mathrm{in}}\right)$. 

So far, the supervised fine adjustment based on MOJS is complete. Firstly, the adaptive learning rate is used to accelerate the unsupervised training process and obtain the initial weight. Secondly, the MOJS algorithm is used to fine-tune the initial weight obtained from the unsupervised process to complete the MOJS-ADBN algorithm process.

## 4. Algorithm and Convergence Analysis

### 4.1. Adaptive Learning Rate CD Algorithm Analysis

(1) Convergence rate refers to the time taken by RBM to use Gibbs sampling many times in order to achieve the expected reconstruction error. The shorter the training time is, the faster the convergence speed is. As a probability model, the unsupervised learning of RBM is mainly used to learn features, which is called the coding adaptive learning rate, which automatically adjusts the learning factor by changing the step size. By comparing the sampling states of the visible layer and the hidden layer every two times, the efficiency of Gibbs sampling improves, and the convergence of the CD algorithm accelerates. Professor Hinton pointed out that hierarchical dimensionality reduction can achieve the effect of exponential reduction in the dimension of high-dimensional data. Similarly, since MOJS-ADBN is a hierarchical representation of multiple RBMs when a single RBM can accelerate convergence through the adaptive learning rate, the convergence speed of DBN will increase exponentially. 

(2) The learning process of RBM weights is different from that of traditional BP networks. RBM is unsupervised learning, while BP is supervised learning; therefore, similar conclusions of the BP algorithm cannot measure RBM. In the unsupervised training stage, the adaptive learning rate algorithm adaptively increases or decreases the learning rate according to the parameter update direction. In addition, in the supervised fine-tuning stage, the algorithm can avoid being in cyclic fluctuations and falling into local optimization in the optimization process. 

At the same time, the adaptive learning rate involves regularly increasing or decreasing the learning intensity of the algorithm on the internal correlation of data in the way of the variable step size, and converging in the shortest time.

### 4.2. Unsupervised Training Phase

In the unsupervised training phase of DBN, to quickly converge, RBM is trained in turn by using the adaptive learning rate. To avoid particularity, in Formulas (4) and (5), the upper and lower asymptotes of the sigmoid function are represented by *A_H_* and *A_L_*, and the input information of the RBM visual layer and the reconstruction state obtained after t samplings are represented by $f_{i}^{0}$ and $f_{j}^{t}$, respectively. Then, the visual layer and hidden layer are expressed as follows in a Gibbs sampling process:(27)$$f_{i}^{0}\in \left[A_{L},A_{H}\right]$$
(28)$$f_{j}^{0}=A_{L}+\left(A_{H}-A_{L}\right)\sigma \left(b_{j}+\sum_{i=1}^{m}f_{i}^{0}W_{ij}\right)$$
(29)$$f_{i}^{1}=A_{L}+\left(A_{H}-A_{L}\right)\sigma \left(a_{i}+\sum_{j=1}^{n}W_{ij}f_{j}^{0}\right)$$
(30)$$f_{j}^{1}=A_{L}+\left(A_{H}-A_{L}\right)\sigma \left(b_{j}+\sum_{i=1}^{m}f_{i}^{1}W_{ij}\right)$$

It can be concluded that, after *t* Gibbs sampling
(31)$$f_{i}^{t}=A_{L}+\left(A_{H}-A_{L}\right)\sigma \left(a_{i}+\sum_{j=1}^{n}W_{ij}^{R}f_{j}^{t-1}\right)$$
(32)$$f_{j}^{t}=A_{L}+\left(A_{H}-A_{L}\right)\sigma \left(b_{j}+\sum_{i=1}^{m}f_{i}^{t}W_{ij}^{R}\right)$$

From the formula, the network output is related to the intermediate state of the sampling process. At the same time, the convergence speed and accuracy of the algorithm are related to the adaptive learning rate. Too large or too small adaptive learning rates will affect the convergence speed and even make the network unstable. From the above, we can obtain the following performance analysis:

(1) Proof of sufficiency.

If $f_{j}^{0},f_{i}^{1}\in \left[A_{L},A_{H}\right]$, according to (27) to (30), then $f_{j}^{1}\in \left[A_{L},A_{H}\right]$. 

(2) Proof of necessity.

On the one hand, if the whole network is stable and the input state of the first RBM satisfies $f_{i}^{0}\in \left[A_{L},A_{H}\right]$, then the output state range of the top RBM satisfies $f_{j}^{1}\in \left[A_{L},A_{H}\right]$, and then it must satisfy $f_{j}^{0},f_{i}^{1}\in \left[A_{L},A_{H}\right]$. 

Proof: 

If the network is stable, the visual and hidden layers of each RBM layer meet the input–output boundedness. Because the sigmoid function is monotonically increasing, and the number of open neurons is also increasing, we can obtain:(33)$$f_{j}^{1}>f_{i}^{1}$$
(34)$$f_{i}^{1}>f_{j}^{0}$$

Then
(35)$$f_{j}^{0},f_{i}^{1},f_{j}^{1}\in \left[A_{L},A_{H}\right]$$

So
(36)$$f_{j}^{t}>f_{i}^{t}$$
(37)$$f_{i}^{t}>f_{j}^{0}$$
(38)$$f_{j}^{0},f_{i}^{t},f_{j}^{t}\in \left[A_{L},A_{H}\right]$$

Furthermore, we know: 

Assume that $f_{j}^{0},f_{i}^{t},f_{j}^{t}$ represent the input state, intermediate state, and output state of RBM, respectively, the sufficient and necessary condition for network stability is:$f_{j}^{0},f_{i}^{t},f_{j}^{t}\in \left[A_{L},A_{H}\right]$. 

According to Formula (6), the greater the δ, the smaller the probability that the neuron takes 1, resulting in the increased sparsity of neurons in the visible layer and hidden layer in the Gibbs sampling process, and the possibility that the weight update direction is the same in the Gibbs sampling iteration process for two consecutive times will increase.
(39)$$P\left\{\;{\left(\Delta w_{ij}^{R}\right)}^{\left(t\right)}+\;{\left(\Delta w_{ij}^{R}\right)}^{\left(t+1\right)}=\;\left|{\left(\Delta w_{ij}^{R}\right)}^{\left(t\right)}\right|\;+\;\left|{\left(\Delta w_{ij}^{R}\right)}^{\left(t+1\right)}\right|\;\right\}\propto \delta$$

According to (8)–(12), if the error fluctuation is not obvious in the process of adjusting the weight, the increase in the learning rate can accelerate the convergence of the weighted network. 

Then there is:(40)$$B\propto P\left\{\;{\left(\Delta w_{ij}^{R}\right)}^{\left(t\right)}+\;{\left(\Delta w_{ij}^{R}\right)}^{\left(t+1\right)}=\;\left|{\left(\Delta w_{ij}^{R}\right)}^{\left(t\right)}\right|\;+\;\left|{\left(\Delta w_{ij}^{R}\right)}^{\left(t+1\right)}\right|\;\right\}$$

According to Gibbs sampling, every time the weight is updated once, the intermediate state is accompanied by two binarization samples, and the updated weight is proportional to the state sampling, so the relationship between *δ* and the learning rate coefficients *B* and *b* can be obtained:(41)$$\left\{ \begin{array}{l} B\approx 2\delta \\ b\approx \delta \end{array}\right.$$

The purpose of *δ* is to judge the state of binary neurons, which is generally 0.7.

### 4.3. Supervised Training Phase

#### 4.3.1. Multi-Objective Jellyfish Behavior Process

For the optimization problem, the calculation formula is as follows:(42)$$\begin{array}{l} \max\;f\left(X\right)\\ s.t.\;g_{i}\left(X\right)\leq 0\;i=0,1,2,\cdots M\;X\in Z \end{array}$$

In the formula, *f*(*X*) is the objective function, *g_i_*(*X*) is the *i*-th constraint, *M* is the total number of constraints, *X* is the n-dimensional unknown variable, and *Z* is the search space. The position state of jellyfish is equivalent to the Pareto optimal solution, and its set represents the Pareto solution set, which is expressed as follows:(43)$$X=\left[X_{1},X_{2},\cdots X_{n}\right]$$

Assuming that the search space *Z* is a continuous state space, the interval $\left[X_{i}^{l},X_{i}^{h}\right]$ where *X* is located can be decomposed into *h*-l discrete values. Then the accuracy can be expressed as $\varepsilon =\frac{X_{i}^{h}-X_{i}^{l}}{h-l}$, in the formula, *ε* is the accuracy of the optimal solution. *Z* is a discrete space, and its state size is:(44)$$\left|Z\right|=\frac{\prod_{i=1}^{n}\left(X_{i}^{h}-X_{i}^{l}\right)}{\varepsilon }$$

The position state X∈Z of each jellyfish, and its food energy, is defined as:(45)F={f(X)|X∈Z}

Then |F|<|Z| is obtained, so:(46)$$F=\{F_{1},F_{2},\cdots ,F_{\left|F\right|}\},F_{1}\text{>}F_{2}\text{>}\cdots \text{>}F_{\left|F\right|}$$

According to the difference of energy, the search space set *Z* can be classified into several non-empty subsets {*Z^i^*}, in the formula:(47)$$Z^{i}=\left\{X|X\in Z,f\left(X\right)=F_{i}\right\}\;i=1,2,\cdots ,\left|F\right|$$

So, $\sum_{i=1}^{\left|F\right|}\left|Z^{i}\right|=\left|Z\right|,$ ∀i∈{1,2,⋯,|F|}, $Z^{i}=\phi ,$ and ∀i≠j,$Z^{i}\cap Z^{j}=\phi$, which satisfy $\overset{\left|F\right|}{\underset{i=1}{\cup }}Z^{i}=Z$. 

The energy of jellyfish (that is food) is defined as:(48)E(X)=f(X)

Let *X_s_* be a set of all jellyfish, *X* is n-vector variable, *X* satisfies $\forall X\in X_{S}$, and $\forall X\in X_{S}$, so $F_{\left|F\right|}\leq E\left(X\right)\leq F_{1}$, set *X_s_* can be reduced to a non-empty subset, and the expression is shown as follows:(49)$$X_{S}^{i}=\left\{X|X\in X_{S},E\left(X\right)=f\left(X\right)=F_{i}\right\}\;i=1,2,3,\cdots ,\left|F\right|$$

So, $\sum_{i=1}^{\left|F\right|}\left|X_{S}^{i}\right|=\left|X_{S}\right|,\forall i\in \left\{1,2,3,\cdots ,\left|F\right|\right\},X_{S}^{i}\neq \phi ,$ and $\forall i\neq j,X_{S}^{i}\cap X_{S}^{j}=\phi$ satisfies $\overset{\left|F\right|}{\underset{i=1}{\cup }}X_{S}^{i}=X_{S}$. 

Let $X^{i,j}$ satisfy $i=1,2,\cdots ,\left|F\right|,j=1,2,\cdots ,\left|X_{S}^{i}\right|$. $X^{i,j}$ represents the position of the *j*-th jellyfish in *X^i^*. Multi-mechanism jellyfish include ocean current movement, jellyfish A-type movement, and jellyfish B-type movement. Assume that the transition of *j*-th jellyfish from one motion state to another is represented by $X^{i,j}\to X^{m,n}$, and the probability of occurrence is $P_{ij,mn}$, assume that the transition of the *j*-th jellyfish from the *i*-th region to the m region in *X^i^* represents $X^{i,j}\to X^{m}$, and the probability of occurrence is $P_{ij,m}$, and satisfies $P_{ij,m}=\sum_{n=1}^{\left|X_{S}^{k}\right|}P_{ij,mn}$, $\sum_{k=1}^{\left|F\right|}P_{ij,m}=1$. Assume that the jellyfish in *X^i^* changes from the *i*-th region to the *m*-th region, indicating $X^{i}\to X^{m}$, and the probability of occurrence is $P_{i,m}$ and satisfies $P_{i,m}\geq P_{ij,m}$.

#### 4.3.2. Stability of Reducible Random Matrix

Theorem 1: Let P be a reducible random matrix of order N, after the same row transformation and column transformation, $P=\left[\begin{array}{l} C\cdots 0\\ R\cdots T \end{array}\right]$, in the formula, *C* is a primitive random matrix of order *M*, *R* and *T* are matrices of order *N-M*, and neither *R* nor *T* is a matrix of **0**. Therefore,
(50)$$P^{\infty }=\underset{k\to \infty }{\lim}P^{k}=\underset{k\to \infty }{\lim}\left[\begin{array}{ccc} C^{k} & \cdots & \;0\\ \sum_{i=1}^{k-1}T^{i}RC^{k-i} & \cdots & \;T^{k} \end{array}\right]=\left[\begin{array}{l} C^{\infty }\;\cdots \;0\\ R^{\infty }\;\cdots \;T \end{array}\right]$$

In the formula, $P^{\infty }$ is a stable random matrix, and $P^{\infty }=1^{\prime }P^{\infty }$, $P^{\infty }=P^{0}P^{\infty }$ are uniquely determined and independent of the initial distribution, $P^{\infty }$ satisfies the condition:(51)$$P^{\infty }={\left[P_{ij}\right]}_{N\times N}=\left\{ \begin{array}{l} P_{ij}>0\;1\leq i\leq N,\;1\leq j\leq M\\ P_{ij}=0\;1\leq i\leq N,\;M<j\leq N \end{array}\right.$$

#### 4.3.3. Proof of Global Convergence

Lemma: in the multi-mechanism jellyfish algorithm, $\forall X^{i,j}\in X_{S}^{i},\;i=1,2,\cdots \left|F\right|,\;j=1,2,\cdots ,\left|X_{S}^{i}\right|$ satisfy:(52)$$\forall m>i,P_{i,m}=0$$
(53)$$\exists m<i,P_{i,m}>0$$here is the proof of Formula (52).

Let $X^{i,j}$ be the artificial jellyfish after *t* iterations, and record it as X(t), the jellyfish with the highest energy in *X*(*t*) is $X_{B\mathrm{est}}$. In the formula, and $X_{B\mathrm{est}}$ is the n-dimensional vector, then there is $E\left(X_{B\mathrm{est}}\right)=F_{i}$. According to the definition of the update archive in multi-mechanism jellyfish, the highest energy jellyfish update in the iteration process can be known as:(54)E(X(t+1))≥E(X(t))

Then
(55)$$\forall m>i,P_{ij,mn}=0$$
(56)$$\forall m>i,P_{ij,m}=\sum_{n=1}^{\left|X_{S}^{k}\right|}P_{ij,mn}=0$$

So
(57)$$\forall m>i,P_{i,m}=0$$here is the proof of Formula (53).

According to the change of time state and environmental state, there will be ocean current movement, jellyfish A-type movement, and jellyfish B-type movement. If *X*(*t* + 1) is the best jellyfish and B(t+1)=X(t+1), the following three phenomena will occur.

Phenomenon 1. Let the jellyfish carry out ocean current movement, and let the probability of producing the ocean current movement be $P_{Ocean}\geq 0$, then the jellyfish group will be attracted by the nutrients in the ocean current to update its position. Then the food concentration at the position before moving is lower than that at the position after moving; that is, E(X(t+1))>E(X(t)), which proves that $\exists m<i,P_{i,m}>0$.

Phenomenon 2. If the jellyfish carries out jellyfish A-type movement, set the probability of generating the jellyfish A-type movement as $P_{A}\geq 0$, and it will move around its own position. Then two situations will occur.

Situation 1. The food concentration at the position after moving is higher than that at the position before moving. Let the probability of this phenomenon be $P_{A1}$, which proves to be the same as Phenomenon 1. 

Situation 2. The food concentration at the current location of the jellyfish is higher than that at the surrounding location. If the probability of this situation is $P_{A2}=1-P_{A1},$ the surrounding location needs to be re-selected. Assume *t* attempts, the probability is $P_{A2}^{t}$. If the food concentration in the position after moving is higher than that in the position before moving, it is the same as that in situation 1. Therefore, if it is still not satisfied after t iterations, according to the time control function *c*(*t*), the jellyfish movement gradually changes from the A-type movement to B-type movement with the increase of times, as shown in Phenomenon 3.

Phenomenon 3. If jellyfish carries out jellyfish B-type movement, it is caused by two conditions.

Situation 1. Jellyfish are produced at the beginning. Let the probability of producing jellyfish B-type movement be $P_{B1}\geq 0$ and $P_{B2}=1-P_{Ocean}-P_{A}$, if the food concentration at the location of a jellyfish in the neighborhood is higher than the food concentration at the current set location, so E(X(t+1))>E(X(t)), which means $\exists m<i,P_{i,m}$.

Situation 2. The jellyfish gradually evolves from A-type movement to B-type movement with the time control function c(t); assume that the probability of occurrence is $P_{B2}\geq 0$, if the food concentration at the location of a jellyfish in the neighborhood is higher than the food concentration at the current set location, then E(X(t+1))>E(X(t)), which means $\exists m<i,P_{i,m}$.

With the increase of *t*, the use of the jump based on opposites can effectively prevent local optimization. 

According to the multi-mechanism jellyfish algorithm, the three movements of jellyfish meet $P_{B2}+P_{Ocean}+P_{A}=1$, and $\exists m<i,P_{i,m}>0$ are proved in each case.

Theorem: 2—the multi-mechanism jellyfish algorithm has convergence.

Proof: $X_{S}^{i},\;i=1,2,\cdots ,\left|F\right|$ is only related to current changes and has nothing to do with history, and the sample space is limited, so it can be regarded as a finite Markov Chain. According to Lemma (1) in Section 4.3.3, the transfer matrix of Markov Chain is:(58)$$P=\left[\begin{array}{cccc} P_{1,1} & 0 & \cdots & 0\\ P_{2,1} & P_{2,2} & \cdots & 0\\ \vdots & \vdots & \cdots & \vdots \\ P_{\left|F\right|,1} & P_{\left|F\right|,2} & \cdots & P_{\left|F\right|,\left|F\right|} \end{array}\right]=\left[\begin{array}{cc} C & 0\\ R & T \end{array}\right]$$

According to Lemma (2) in Section 4.3.3:(59)$$P_{2,1}>0,\;R={\left(P_{2,1},P_{3,1},\cdots ,P_{\left|F\right|,1}\right)}^{T}$$
(60)$$T=\left[\begin{array}{ccc} P_{2,2} & \cdots & 0\\ \vdots & \cdots & \vdots \\ P_{\left|F\right|,2} & \cdots & P_{\left|F\right|,\left|F\right|} \end{array}\right]\neq 0,\;C=P_{1,1}=1$$

If *P* is a reducible random matrix of order *N*, then

$P^{\infty }=\underset{k\to \infty }{\lim}P^{k}=\underset{k\to \infty }{\lim}\left[\begin{array}{ccc} C^{k} & \cdots & \;0\\ \sum_{i=1}^{k-1}T^{i}RC^{k-i} & \cdots & \;T^{k} \end{array}\right]=\left[\begin{array}{l} C^{\infty }\;\cdots \;0\\ R^{\infty }\;\cdots \;T \end{array}\right]$ and $C^{\infty }\;=1,\;R^{\infty }={\left[1,1,,\cdots ,1\right]}^{T}$.

Therefore, $P^{\infty }=\left[\;\begin{array}{cccc} 1 & 0 & \cdots & 0\\ 1 & 0 & \cdots & 0\\ \vdots & \vdots & \vdots & \vdots \\ 1 & 0 & \cdots & 0 \end{array}\right]$ is a stable random matrix, which leads to:(61)$$\underset{t\to \infty }{\lim}P\left\{E\left(X\left(t\right)\right)=F_{B}\right\}=1$$

In the formula, *F_B_* is the optimal objective function, so the multi-mechanism jellyfish algorithm has global convergence.

#### 4.3.4. Global Stability Proof

From Section 4.3.3, it can be seen that the multi-mechanism jellyfish algorithm finally converges to the global best, so the initial position of *X*_0_ will eventually converge to the global best *x_max_*. *x_max_* is assumed to be the equilibrium point under the Lyapunov meaning. 

Proof: Assume the objective function of the multi-mechanism jellyfish algorithm is *f*(*X*), then the dynamic formula is:(62)$$\dot{X}=f\left(X,t\right)$$

Let the x axis translate $f\left(x_{\max}\right)$ upward, then the dynamic formula is updated as:(63)$$\dot{X}=f\left(X,t\right)-f\left(x_{\max}\right)$$

According to the convergence of algorithm, when t→∞, the position state *X* of jellyfish tends to the global best *x_max_*:(64)$$\underset{t\to \infty }{\lim}\|X\left(t:\;X_{0},t_{0}\right)-X_{e}\|=0$$

So, for all *t*, the equilibrium state is satisfied.
(65)$${\dot{X}}_{e}=f\left(X_{e},t\right)-f\left(x_{\max}\right)=0$$

In the formula, *x_max_* is the equilibrium point in the MOJS algorithm, and ${\dot{X}}_{e}=f\left(x_{\max},t\right)$ is the equilibrium state. Therefore, there are equilibrium points and equilibrium states in the MOJS algorithm.

#### 4.3.5. Stability of the MOJS Algorithm in the Lyapunov Meaning

Assume that the initial condition state of the MOJS algorithm is within the hyper-sphere S(δ) with the equilibrium point *x_max_* as the center and *δ* as the radius, then X∈S(δ) can represent $S\left(\delta \right)=\left\{X|\left\|X-x_{\max}\right\|\leq \delta \right\}$; that is:(66)$$\left\|X-x_{\max}\right\|\leq \delta$$

As shown in Figure 3, S(γ) is a circle with a center radius of γ and a circle S(δ) with a center radius of *δ*, the points on both sides of the side are set as $x_{1},x_{2}$, the circle S(γ) and *f*(*x*) intersect with $x_{3},x_{4},$ assume *f*(*x*) is the objective function graph, *f*(*x_max_*) is the maximum value of the function, *f*(*x_max_*_1_) is the next largest value of the function, and *S_max_* is the region between the maximum value and the next largest value of the objective function, which is called the optimal region.

It is assumed that the MOJS algorithm satisfies the stability in the Lyapunov meaning, and the equilibrium state is uniformly asymptotically stable.

Proof: According to the global convergence of MOJS in Section 4.3.3, when *X* is in *S_max_*, *X* will be attracted by food and move towards *x_max_*, so the initial solution $X\left(t:\;X_{0},t_{0}\right)$ of the equation is located in *S_max_*, and *S_max_* is included in the intersection region of S(δ) and *f*(*x*), then *X* will not escape S(δ). Then *δ* satisfies:(67)$$\left\{ \begin{array}{l} \delta \leq \min\left(\left\|x_{\max}-x_{1}\right\|,\left\|x_{\max}-x_{2}\right\|\right)\\ \delta \leq \min\left(f\left(x_{\max}\right)-f\left(x_{3}\right),f\left(x_{\max}\right)-f\left(x4\right)\right) \end{array}\right.$$

So
(68)$$\|X\left(t:\;X_{0},t_{0}\right)-X_{e}\|\leq \gamma ,\;t\geq t_{0}$$

Therefore, when t→∞, ∀t,  makes $X\left(t:\;X_{0},t_{0}\right)\in S\left(\gamma \right)$, it satisfies the stability under the Lyapunov meaning.

If ∀γ > 0, ∃δ and *δ* satisfy formula (67), and the initial state *x*_0_ satisfies $\left\|x_{0}-x_{\max}\right\|\leq \delta ,$ then *x*_0_ satisfies $∥X(t:X_{0},t_{0})\;-\;X_{e}∥\;\leq \;\delta$. Therefore, it can be concluded that *δ* is independent of *t*_0_, and the equilibrium state *x_max_* of the MOJS algorithm is uniformly stable, which is proved.

So far, the stability proof of the membrane fouling fault diagnosis model based on MOJS-ADBN has been completed.

## 5. Simulation Experiment and Research Analysis

### 5.1. Membrane Fouling Data Acquisition

We used CFD software aimed at the problem that the membrane flux is easily affected by influent flow and temperature; this article used the parallel hollow fiber membrane device as the research object, and accurately classified the factors that cause membrane pollution. CFD software was used to simulate and calculate the water production in the MBR system to collect fault data. 

Using the modeling process of the parallel hollow fiber membrane unit as an example, the Euler multiphase flow model was selected to simulate and build the MBR simulation system. The equation of mass and momentum conservation is as follows.

Mass-conservation equation:(69)$$\frac{\partial }{\partial t}(\alpha_{q}\rho_{q})+\nabla \cdot (\alpha_{q}\rho_{q}\mu_{q})=0$$

In the formula, $\alpha_{q}$ is volume, $\rho_{q}$ is density ($\mathrm{kg}\;\cdot {\;m}^{-3}$), $\mu_{q}$ is he average velocity vector of *q*-th ($m\;\cdot {\;s}^{-1}$), and *q* is liquid *s*.

Momentum conservation equation:(70)$$\frac{\partial \alpha_{q}\rho_{q}\mu_{q}}{\partial t}+\nabla g(\alpha_{q}\rho_{q}\mu_{qj}\mu_{q})=-\alpha_{q}g\nabla p_{q}+\nabla g(\alpha_{q}\tau_{q})+F_{q}+\alpha_{q}\rho_{q}g$$

In the formula, *q* represents the liquid phase, *j* represents x, y, z in three directions, $\alpha_{q}$ is volume fraction, $\mu_{q}$ is velocity ($m\;\cdot {\;s}^{-1}$), $\rho_{q}$ is density ($\mathrm{kg}\;\cdot {\;m}^{-3}$), *P_q_* is pressure (Pa), $\tau_{q}$ is viscous stress tensor (Pa), *F_q_* is interaction force ($N\;\cdot {\;m}^{-3}$), *g* is gravitational acceleration ($m\;\cdot {\;s}^{-2}$).

In the control of the solution, we set up the solution method at first. In the drop-down list of pressure–speed coupling, a phase-based coupling algorithm was selected to calculate the grid file. In the differential discrete format option, we set the gradient to cell-based least squares and the transient item format to first-order implicit. We set the monitoring window and convergence threshold. In the simulation data, we summarized nine types of membrane contamination data, such as too large, too small, and within the tolerance range; these data were collected for the main influencing factors of membrane contamination.

According to the analysis of the importance of membrane pollution factors, when the transmembrane pressure difference was constant, the above four influencing factors were selected as the research objects for analysis because the concentration difference of COD in and out water (*C*), BOD in and out water (*B*), solid concentration of mixed suspension (*X*) and hydraulic retention time (*H*) had obvious effects on membrane pollution. After testing and comparison, a tolerance of 5% was set for the COD concentration difference and BOD concentration difference of the inlet and outlet water in the series tubular membrane device, and a tolerance of 7% was set for the mixed suspension solid concentration and hydraulic retention time. A tolerance of 5% was set for the values of the membrane fouling factors in the parallel hollow fiber membrane device. When the membrane fouling factor value was within the set tolerance range, it indicated that there was no pollution in the series tubular membrane device. When the values of the membrane pollution factors exceeded the set tolerance, it meant that membrane pollution factors, such as COD concentration difference, BOD concentration difference, mixed suspension solid concentration, and hydraulic retention time were too large, resulting in membrane pollution. The types of membrane pollution are f2, f4, f6, and f8, respectively When the value of the membrane pollution factor was lower than the set tolerance, it indicated that the COD concentration difference, BOD concentration difference, mixed suspension solid concentration, and hydraulic retention time of the inlet and outlet water were too small, resulting in membrane pollution. The categories of membrane pollution are f3, f5, f7, and f9. Membrane pollution codes f1–f9 correspond to different membrane pollution types caused by “normal”, “too large”, and “too small” membrane pollution factors of the parallel hollow fiber membrane device in the actual operation of the membrane water treatment; see Table 1.

To better speed up the training of the network model, we made the data easy to calculate, obtained more generalized results, and the input data were standardized; the mathematical expression is:(71)$$X=\frac{X-X_{\min}}{X_{\max}-X_{\min}}$$

### 5.2. Experimental Process

The experimental processes of this article are fault data collection, fault classification and coding, data pre-processing, data analysis and division, MOJS-ADBN model construction, prediction coding, and result analysis. The specific steps are as follows:

(1) Take membrane fouling data.

(2) Encode the data classification of membrane fouling.

(3) Classify the data into a training set and test set according to the ratio of 7:3.

(4) Build the MOJS-ADBN model, retain the weight in the unsupervised learning process, and use the adaptive learning rate to accelerate the training process. In the process of supervised learning, MOJS is used to optimize the algorithm and fine-tune the weight. The training set is used to adjust the network model to make the model optimal.

(5) Compare the actual code of the test set with the prediction code generated by the model. If the prediction code is consistent with the real coding result, the classification is correct; if the prediction code is inconsistent with the real coding result, the classification is wrong.

(6) Further analyze the model and judge the performance of the model from the perspective of average accuracy, average precision, average recall, and running time.

In this article, the MOJS-ADBN hidden layer was set as three layers, and the optimal number of hidden layer neurons was selected to determine the optimal number of hidden layer neurons based on the model error and running time. According to the experimental method, when the number of neurons in the hidden layer is 20, the performance effect is the best, as shown in the Figure 4. At this time, ape and MSE are 0.0618 and 0.0742, respectively. Figure 4 shows the relationship between the model error and the number of hidden layer neurons. In the formula, ape and MSE represent the absolute percentage error and mean square error, respectively.
(72)$$\mathrm{APE}=\frac{1}{N_{t}}\sum_{i=1}^{N_{t}}\left|\frac{{\hat{y}}_{i}-y_{i}}{y_{i}}\right|\times 100\%$$
(73)$$\mathrm{MSE}=\frac{1}{N_{t}}\sum_{i=1}^{N_{t}}{\left({\hat{y}}_{i}-y_{i}\right)}^{2}$$

In the formula, *y_i_* and ${\hat{y}}_{i}$ represent the real value and predicted value, respectively, and *N_t_* represents the number of test samples.

To objectively prove that the best model structure of MOJS-ADBN is 18-20-20-20-9, 300 data were collected for each membrane pollution category of the parallel hollow fiber membrane device, with a total of 2700 experimental data. A total of 1890 samples were randomly selected as training samples, and the remaining 810 samples were used as test samples.

In the unsupervised training phase, each RBM was set to iterate 378 times, and the learning rate coefficients were set to B = 1.4 and b = 0.7, respectively; the kernel principal component analysis (KPCA) was used to extract the three principal components of the first RBM output feature and the three principal components of the final DBN output feature, which are represented in Figure 5a,b respectively. It can be seen from Figure 5a that only f1 does not overlap with other faults; f2, f7, and f9 overlap, and the distribution of similar faults in f2 and f7 is relatively scattered. Moreover, f4, f5, and f8 overlap seriously, and the fault types cannot be classified correctly. Although f3 and f6 can be classified, there is still a small amount of overlap. It can be seen from Figure 5b that all kinds of faults do not overlap and can be classified better. Therefore, the DBN model can accurately distinguish other fault categories, and the distribution of similar faults is more compact than that in Figure 5a because the input data will undergo (four times) nonlinear mapping and the data will be reconstructed after passing through four RBMs, which can more accurately and abstractly express the input data.

We used the MOJS algorithm for supervised fine adjustment. We set the three layers as in [9, 20, 20, 20], respectively, to establish the MOJS-ADBN model. Figure 6a,c,e,g represent the Pareto front scatter diagram, in the formula, and the abscissa and ordinate represent the objective function of the Pareto optimal solution respectively; while Figure 6b,d,f,h represent the Pareto frontier broken line graph, in the formula, the abscissa represents the number of Pareto optimal solutions, the two broken lines represent the objective functions of the Pareto optimal solutions, respectively, and the color block in the graph represents the overlapping part of the Pareto frontier scatter diagram. It can be seen from the graph that the weight can be improved after four times the MOJS algorithm optimization and supervised fine-tuning, make the weight distribution more reasonable.

To reduce the influence of experimental randomness on the evaluation of the model diagnostic performance, 10 independent diagnostic experiments were carried out on the parallel hollow fiber membrane device. Figure 7a presents the average confusion matrix of 10 diagnostic faults of the MOJS-ADBN model. From the figure, it can be seen that there are 9 fault codes from f1 to f9, and each fault is counted 900 times in total. In the formula, the total number of f1 misclassifications is 14. In the formula, misclassification is: f2—five times, f6—three times, f9—two times, and f3, f4, f7, and f8 are misclassified once each; the total number of false divisions is 14; f1 is classified six times, f6 is classified four times, and f3, f5, f8, and f9 are classified once each. The total number of f3 misclassifications is eight; f1, f4, f7, f9 are misclassified once each, and f5 and f8 are misclassified twice each. The total number of f4 misclassifications is 8; f1, f2, and f7 are misclassified once, f5 is misclassified three times, and f8—twice. The total number of misclassifications is 20. In the formula, misclassification is f1—five times, f2—eight times, exception misclassifications of f3, f4, f5, f7, and f8—once each, misclassification of f9—twice. The total number of f7 misclassifications is 9; f1, f3, f5, and f9 are misclassified once each, f4 is misclassified three times, f8 is misclassified two times. The total number of f8 misclassifications is 8; f1, f3, f6, and f7 are misclassified once each, f4 and f5 are misclassified twice. The total number of f9 misclassifications is eight. In the formula, misclassifications of f5, f6, and f8 are once each, and misclassifications of f1 and f2 are two times each. From the figure, it can be seen that f1, f2, and f6 are easy to be confused compared with the other faults. Figure 7b shows the curve of the accuracy, accuracy, and recall of all kinds of faults. From the figure, it can be seen that the accuracy, accuracy, and recall of all kinds of faults is above 97%; therefore, the MOJS-ADBN proposed in this article still has strong robustness.

### 5.3. Comparative Test

#### 5.3.1. Comparative Test of Different Learning Rates

As a probability model, RBM is mainly affected by weight, so a reasonable weight is the premise to ensure accurate network classification. Figure 8 shows the weights obtained by using the adaptive learning rate and fixed learning rate, respectively. As can be seen from the figure below, the weight distribution obtained by using the adaptive learning rate is more compact than that obtained by the fixed learning rate, which can effectively avoid the problems of ignoring detailed features or gradient disappearance caused by too large or too small weights.

In the past, the learning rate of the DBN was determined by experience. To further prove the applicability of the adaptive learning rate, comparative experiments were used to verify 0.01, 0.05, 0.1, 0.5, and 1 as the learning rates of RBM, the supervised learning part was fixed, and 10 experiments were carried out on the parallel hollow fiber membrane device as the research object. The training and test data were classified for verification. Table 2 shows the diagnostic comparison experiment. It can be seen from the table that the diagnostic accuracies of learning rates 0.1 and 1 were higher than that of other learning rates, but the adaptive learning rate proposed in this article not only ensured the accuracy but also accelerated the network convergence. Therefore, the adaptive learning rate proposed in this article, based on the setting of the parameter update direction, progressed compared to the traditional empirical method.

#### 5.3.2. Comparison of Ablation Experiments

To further prove the effectiveness and superiority of the MOJS-ADBN model for membrane device–membrane fouling diagnosis, this method is compared with some common fault diagnoses and classification methods. We combined wavelet transform with PCA to extract features. In the learning of the shallow neural network, BP, extreme learning machine (ELM), SVM, and least square support vector machines (LSSVM) were used for classification diagnosis. In deep learning, the traditional DBN and adaptive learning rate DBN (ALRDBN) were used, the data set was expanded by overlapping sampling, and then the convolutional neural network (CNN) was used for comparison. According to the method in this article, training data and test data were classified for 10 independent diagnosis experiments, and the comparison indicators included network structure, average time, mean value, and variance of the test MSE; the results are shown in Table 3. It can be seen from the table that compared with the shallow network, the DBN can effectively extract the essence and depth characteristics of faults. After optimization, the DBN improved both the accuracy and network performance to varying degrees, and the nonlinear mapping between the initial data and characteristics were more obvious. Compared with the deep network, although the CNN has a lower diagnosis time than MOJS-ADBN, the diagnosis rate of the improved CNN is lower than MOJS-ADBN, and the CNN needs a large number of data sets and reasonable division to ensure the rationality of the model, so the MOJS-ADBN proposed in this article is more conducive to the accurate identification of faults.

The parallel hollow fiber membrane device membrane fouling simulation data set was used to carry out ablation experiments. Five performances, including average accuracy, average accuracy, average recall, average time, and average determination coefficient R^2^ were used as the bases for the model judgment, and the performances of the DBN ALRDBN improved CNN, and MOJS-ADBN were verified, respectively.

According to the analysis in Figure 9, the performance of the improved model in this article improved to varying degrees. Although the reduction effect of the running time was not prominent, the accuracy significantly improved (besides the running time), while the other four performance effects of the MOJS-ADBN model were significantly better than the other three network models, which verifies the effectiveness and superiority of the MOJS-ADBN diagnostic model proposed in this article. 

#### 5.3.3. Variable Noise Membrane Fouling Diagnosis Results of Different Diagnostic Methods

During the actual operation of the membrane bioreactor, there was environmental noise when the membrane component was treating sewage. At the same time, due to the characteristics of the membrane component itself, there was also noise, which produced unnecessary randomness in the collection of the membrane pollution data. At the same time, because the simulated data needed to be more consistent with the uncertainty of the operation of the membrane component under the actual working conditions, it was very important to add the variable noise experiment to the membrane fouling diagnosis experiment. To verify whether this method could obtain higher fault diagnosis accuracy and better generalization ability in the variable noise experiment, the experimental results of this article were compared with the experimental results of the methods proposed in references [34] and [36]. Reference [34] proposed a DBN fault diagnosis model with an improved model structure. The model uses multi-layer and multi-dimensional mapping to extract more detailed fault type differences and accurately diagnose faults. Reference [36] used the model after optimizing the parameters of a support vector machine to diagnose the signal features extracted by the DBN, realized the online detection of equipment faults, and improved the accuracy of the diagnosis. In this article, aimed at the membrane pollution data of the parallel hollow fiber membrane component as the training sample, Gaussian white noise (with SNRs of −2, 0, 2, and 4 dB) was added to the test sample, and the obtained membrane fouling diagnosis results were compared with other diagnostic methods. The experimental results are shown in Table 4

From Table 4, it can be seen (from the comparative data of four) that in the experimental results of different SNRs, the accuracy of the membrane component-membrane fouling diagnosis based on MOJS-ADBN was higher than that of other methods, and its anti-noise performance was stronger than the first three diagnostic methods.

## 6. Conclusions

This article presents a method of membrane packaging diagnosis based on MOJS-ADBN to optimize the DBN from the perspectives of unsupervised learning and supervised learning:

(1) The adaptive learning rate was used to accelerate the convergence of the network and proved that the unsupervised part optimized by the adaptive learning rate was stable. 

(2) The supervised part used the MOJS algorithm optimization to fine-tune the weight, proving that MOJS optimization has global convergence and stability in the Lyapunov meaning. 

(3) MOJS-ADBN was verified by a simulation experiment with a parallel hollow fiber membrane component. The experimental results show that the MOJS-ADBN model can effectively classify and locate faults, and can be used as a new solution in the field of membrane fouling diagnosis for membrane water treatment.

## Figures and Tables

**Figure 1 membranes-12-00843-f001:**
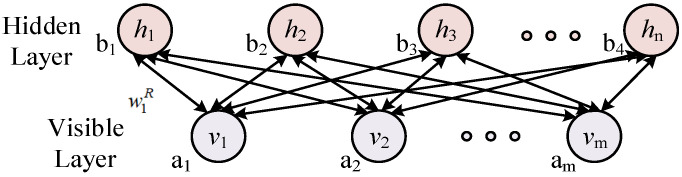
Structure of the RBM.

**Figure 2 membranes-12-00843-f002:**
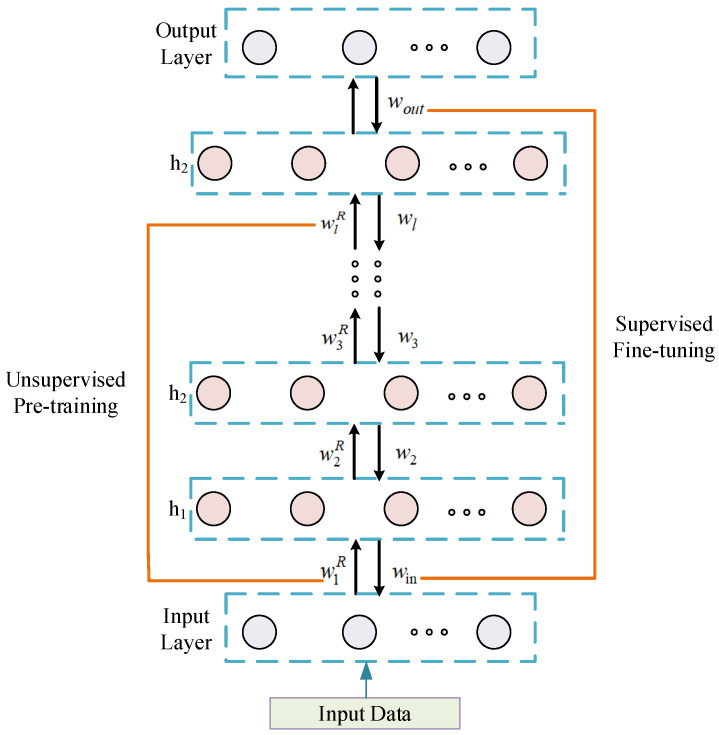
Structure of DBN.

**Figure 3 membranes-12-00843-f003:**
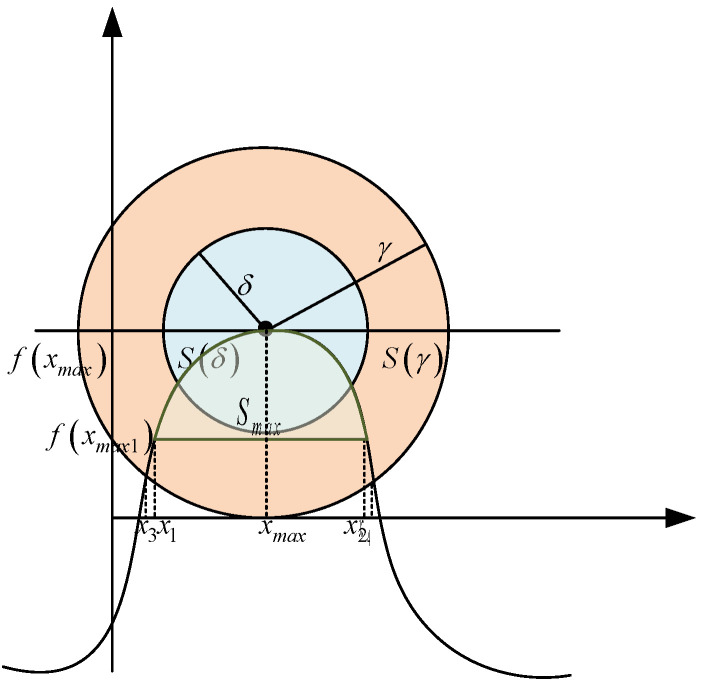
Stability of the MOJS algorithm in the Lyapunov meaning.

**Figure 4 membranes-12-00843-f004:**
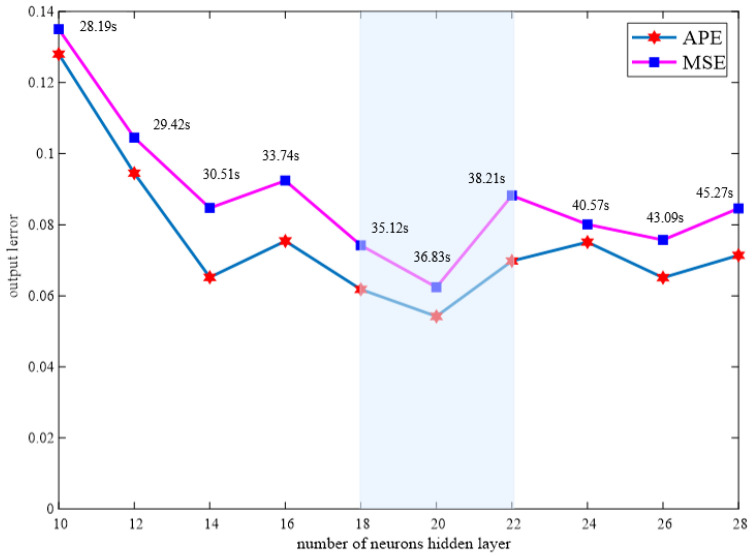
Relationship between the MOJS-ADBN model error and the number of hidden layer neurons.

**Figure 5 membranes-12-00843-f005:**
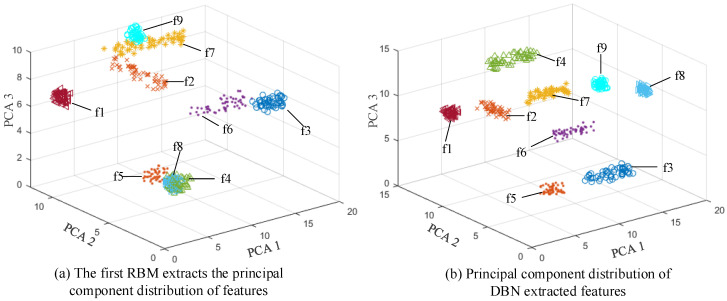
Principal component distribution of feature extraction.

**Figure 6 membranes-12-00843-f006:**
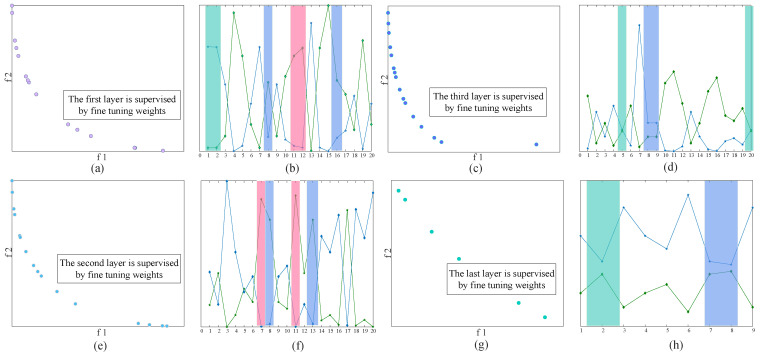
Pareto frontier analysis of the MOJS-ADBN model.

**Figure 7 membranes-12-00843-f007:**
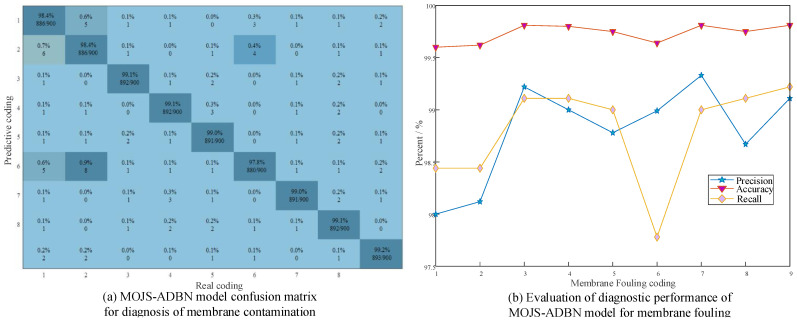
Performance of the MOJS-ADBN model in membrane fouling diagnosis.

**Figure 8 membranes-12-00843-f008:**
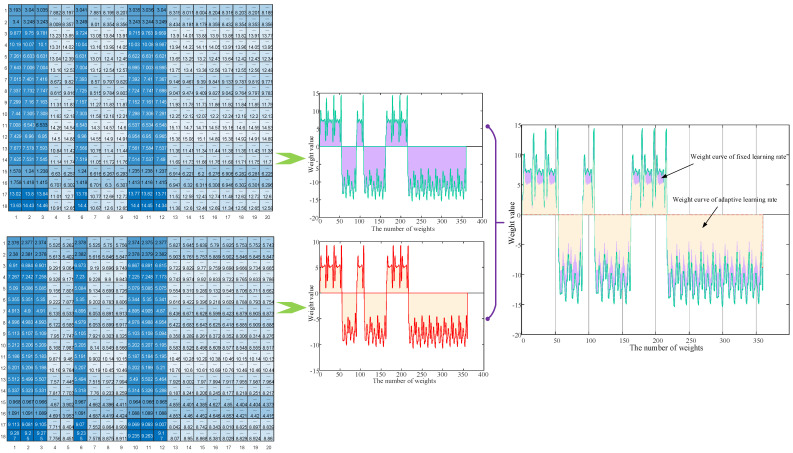
Comparison of optimization weights of different learning rates.

**Figure 9 membranes-12-00843-f009:**
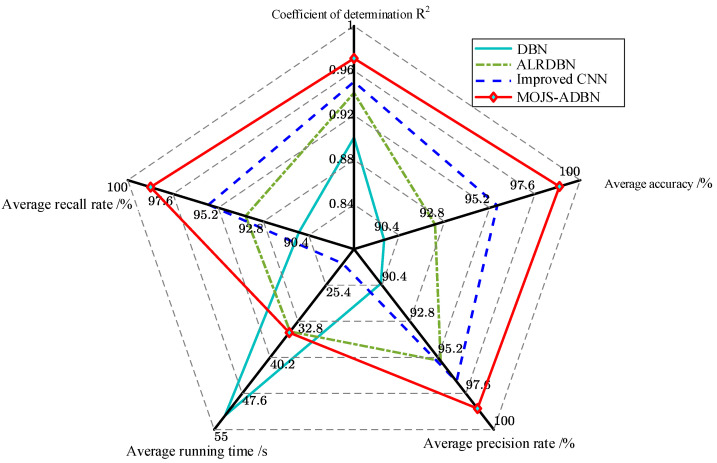
Performance comparison results of the ablation experiments.

**Table 1 membranes-12-00843-t001:** Membrane fouling mode of the membrane device.

Fault Code	Fault Type	Tolerance
f1	No fouling	—
f2	C too large	5%
f3	C too small	5%
f4	B too large	5%
f5	B too small	5%
f6	X too large	7%
f7	X too small	7%
f8	H too large	7%
f9	H too small	7%

**Table 2 membranes-12-00843-t002:** Diagnostic accuracies of different fixed learning rates.

Learning Rate	Average Accuracy/%
0.01	95.26
0.05	93.73
0.1	96.21
0.5	94.57
1	96.75

**Table 3 membranes-12-00843-t003:** Comparison of diagnostic performances of different models.

Diagnosis Method	Network Structure	Testing MSE		Average Time/s	Average Accuracy/%
		Mean	Variance		
BP	18-20-9	0.0294	0.0121	55.42	78.51
ELM	18-20-9	0.0313	0.0106	59.47	81.05
SVM	Gaussian Kernel Function	0.0251	0.0092	62.73	80.93
LSSVM	Gaussian Kernel Function	0.0247	0.0085	60.51	83.57
DBN	18-20-20-20-9	0.0218	0.0075	52.14	90.92
ALRDBN	18-20-20-20-9	0.0157	0.0053	34.91	93.75
Improved CNN	21 layers	0062	0.0035	20.97	95.72
MOJS-ADBN	18-20-20-20-9	0.0052	0.0027	35.12	98.79

**Table 4 membranes-12-00843-t004:** Diagnosis accuracy rates of different methods under different noises.

Diagnostic Method	SNR/dB			
	−2	0	2	4
DBN	87.17%	91.08%	89.38%	90.74%
Reference [34]	94.11%	96.20%	96.03%	95.77%
Reference [36]	94.21%	95.97%	96.12%	96.33%
MOJS-ADBN	96.42%	98.94%	98.16%	98.23%

## Data Availability

Data is contained within the article.

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
