# Peer review of "Membrane Fouling Diagnosis of Membrane Components Based on MOJS-ADBN"

_membranes, 2022, doi:10.3390/membranes12090843_

Round 1

Reviewer 1 Report

The authors propose and test an approach to detection of factors affecting fouling of membranes in MBR processes.  The presentation of the method is extensive while the description of the dataset used to test the proposed method is quite deficient.  The source and extent of data used to test the method needs to be described in sufficient detail to allow readers to judge whether the dataset is sufficient to allow reasonable testing to be conducted.

Author Response

Thank you very much for your advices and comments of my work. We also agree with your suggestion and have carefully supplemented and revised them in accordance with your suggestion. According to your suggestion, we have supplemented the source and extent of the data used to test the method proposed in this paper in the manuscript, and the modified parts have been marked in red.

Reviewer 2 Report

The authors have reported algorithm of data-processing for membrane packing diagnostics based on multi objective jellyfish search adaptive deep belief network. Due to high efficiency of the obtained results from the application of the model shown in Table 4; and its potential use to practice, the paper would be of interest in the readers of Membrane.

Author Response

Thank you very much for your confirmation of our manuscript.
